# Crustaceans in the Meiobenthos and Plankton of the Thermokarst Lakes and Polygonal Ponds in the Lena River Delta (Northern Yakutia, Russia): Species Composition and Factors Regulating Assemblage Structures

**Elena S. Chertoprud** [1,2,*] **and Anna A. Novichkova** [1,2,3]

1   Department of Main Ecology and Hydrobiology, Biological Faculty, Moscow State University, Leninskie Gory, 119991 Moscow, Russia; anna.hydro@gmail.com
2   Severtsov Institute of Ecology & Evolution, Leninsky Pr. 33, 119071 Moscow, Russia
3   State Nature Reserve "Wrangel Island", Kuvaeva Str. 23, Chukotka Autonomous Region, 689400 Pevek, Russia
*   Correspondence: horsax@yandex.ru

**Abstract:** Information about invertebrates in the low-flow water bodies of northeastern Siberia is far from complete. In particular, little is known about crustaceans—one of the main components of meiobenthic and zooplanktonic communities. An open question is which environmental factors significantly affect the crustaceans in different taxonomic and ecological groups? Based on the data collected on the zooplankton and meiobenthos in the tundra ponds in the southern part of the Lena River Delta, analysis of the crustacean taxocene structure was performed. In total, 59 crustacean species and taxa were found. Five of these are new for the region. The species richness was higher in the large thermokarst lakes than in the small water bodies, and the abundance was higher in small polygonal ponds than in the other water bodies. Variations in the Cladocera assemblages were mainly affected by the annual differences in the water temperature; non-harpacticoid copepods were generally determined by hydrochemical factors; and for Harpacticoida, the macrophyte composition was significant. Three types of the crustacean assemblages characteristic of different stages of tundra lake development were distinguished. The hypothesis that the formation of crustacean taxocenes in the Lena River Delta is mainly determined by two types of ecological filters, temperature and local features of the water body, was confirmed.

**Keywords:** Lena River Delta; zooplankton; meiobenthos; Cladocera; Copepoda; ecological factors





## 1. Introduction

The Lena River Delta, which has an area of approximately 30,000 km², is the largest delta in the Arctic and third largest in the world after the Amazon and Ganges deltas [1]. In its territory, dissected by numerous channels, there are more than 30 thousand lakes, polygonal, thermocarst or oxbow in origin, most of which are small (less than 0.25 km²). Fluvial river sediments in the delta formed river terraces enclosing three levels in the Late Pleistocene and during the Holocene [2]. The different geneses, deposit types, and relief-forming processes of these geomorphological terraces caused the differences in their predominant lake types [3].

The first data collected on freshwater crustaceans (primarily Cladocera and Copepoda) of the Lena River Delta were obtained during the Russian Polar Expedition of 1901–1903 and presented by Rylov [4] and Behning [5]. Based on these data, a general list of fauna, including approximately 50 species and forms, was compiled. Later, while analysing materials collected in the Lena River and delta channels, as well as in Tiksi, Oleneksky and Neelova bays, a more complete list of planktonic organisms was compiled, merging 75 taxa [6]. From the vicinity of Tiksi Bay, an amphipod species from the Crangonyctidae

family Bousfield, the 1973 endemic to the region was described [7]. Information on most zooplankton and meiobenthos of the Lena Delta has been obtained from only the largest river channels [6,8–11], while small reservoirs of floodplain terraces have remained disregarded for a long time. However, it was noted that in comparison to those in low-flow reservoirs (lakes and oxbows), planktonic communities in rivers are inferior in terms of species richness and abundance [12].

Until recently, there was almost no information about the composition of crustacean communities in the thermokarst lakes and polygonal ponds of the Lena River Delta. In recent years, a number of studies have been conducted on the zooplanktonic communities of Cladocera and Copepoda of the lakes of the southern part of the delta, primarily Samoilovsky Island [13–17]. Research on microcrustaceans has become particularly relevant in the context that biocenoses of polygonal tundra are considered to be sensitive to environmental and climate changes [18]. Zooplanktonic species of the polygonal ponds and thermokarst lakes are good proxies of climate and environmental changes and therefore may be subject to monitoring studies [19].

The effect of river runoff on the composition of the Cladocera and Copepoda of oxbows and lakes for 2000–2015 was analysed. The results showed that the core of the fauna is composed of indigenous Arctic species, but the fauna also includes invasive taxa from temperate latitudes and relics of the ice age [13,15]. Biotopic variability and seasonal dynamics of the taxocene structure of planktonic crustaceans in polygonal ponds, thermokarst lakes and old lakes have been described [14–17]. Ecological assemblages of Copepoda characteristic of different types of water bodies were allocated [20]. Recently, the first integrated ecological study on the fauna of the benthic Copepoda of the order Harpacticoida in different waterbodies of the delta was conducted [21]. A comparative analysis of the structure of the Cladocera and Copepoda assemblages, typical for the benthic and planktonic communities of the polygonal ponds and thermokarst lakes, has not been previously conducted. It is not obvious which environmental factors are critical for formation of the assemblages of different taxonomic and ecological (plankton and meiobenthos) groups within a single reservoir in the Arctic.

The current research analyses is focused on the taxocene structure of the crustaceans inhabiting the bottom sediments and the water depths of the polygonal ponds and thermokarst lakes on the two neighbouring islands of Kurungnakh and Argaa-Bilir-Aryata in the southern part of the Lena River Delta. Taxonomic composition is described, abundance is estimated, and the key environmental factors determining the variability in the assemblages of crustaceans in the plankton and meiobenthos in the summer period are identified. The novelty and the value of this study lie in the simultaneous analysis of many taxonomic and ecological (benthos and plankton) groups of crustaceans. At the same time, freshwater ecosystems of high latitudes are key sites for monitoring and observation of global climate change.

## 2. Materials and Methods

### 2.1. Studied Area

Studies were performed during the summer seasons (July and August) of 2017 and 2020 on the islands of Kurungnakh and Argaa-Bilir-Aryata, located in the southern part of the Lena River Delta (Figure 1a,b). This area is characterized by an Arctic continental climate with an average annual temperature of approximately −14 °C and average annual precipitation of 125–190 mm [22,23]. The winter season (average temperature −30 °C) lasts for six months, from the end of September to the end of March. The summer period, which is characterized by relatively high temperatures (on average 7 °C and maximum 20 °C) and constant lighting, lasts for two months [23]. Permafrost depth averages 30–50 cm [1].

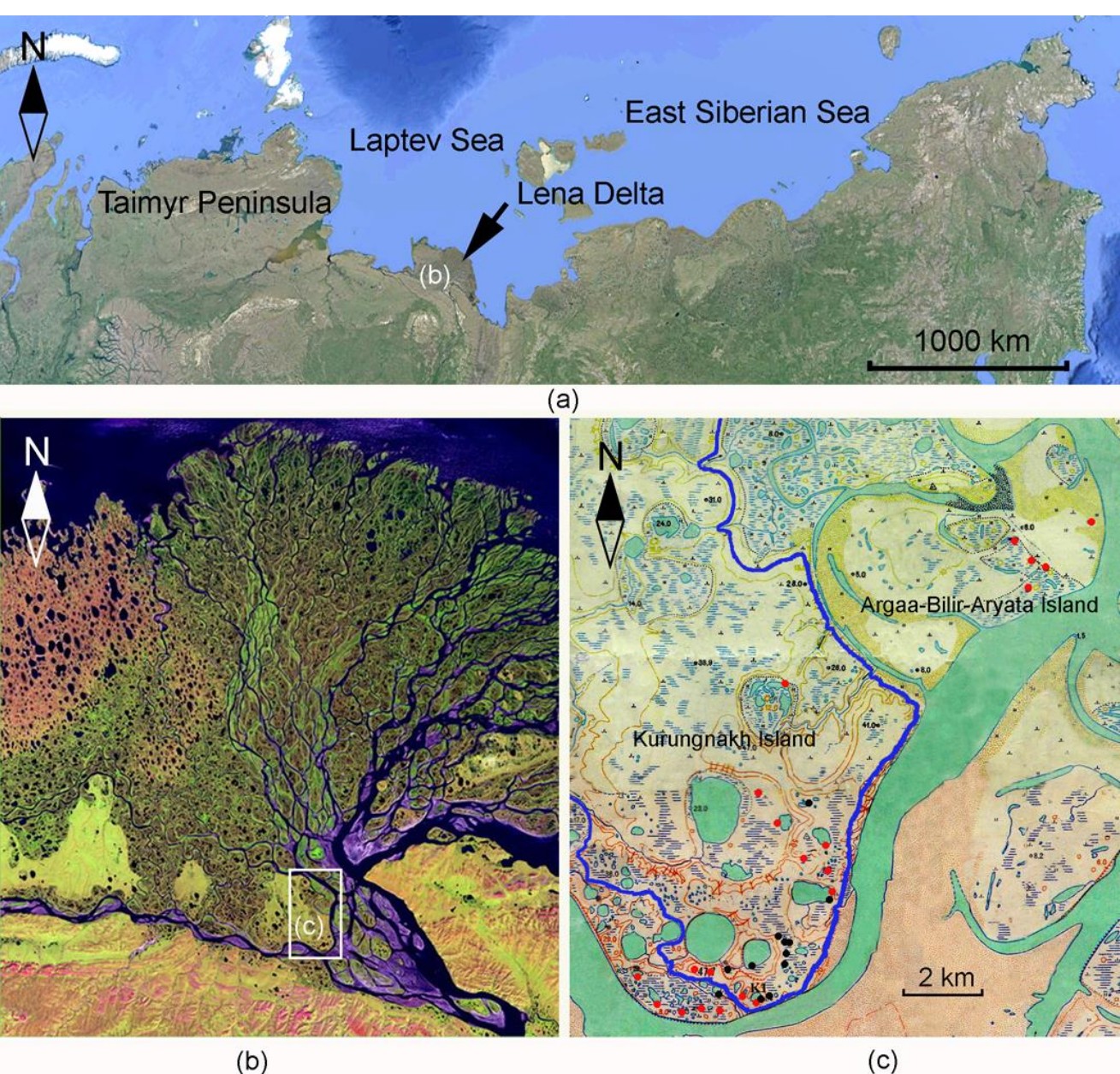

**Figure 1.** Map of Eastern Siberia (**a**) with position of the Lena Delta (arrow); Lena River Delta (**b**) with location of sampling area (white square); sampling area (**c**) with positions of sampling points (red2017, black2020) and border between first and second river terraces (blue line).

The investigated islands are covered with typical moist sedge-moss tundra communities [24]. In addition, the southwestern and northern extremities of Argaa-Bilir-Aryata Island have extensive silty-sandy bars that flood during the spring. Kurungnakh Island includes the areas related to both the first and the third river terraces, while Argaa-Bilir-Aryata Island includes areas only related to the first terrace (Figure 1c) [25]. The first terrace features a high density of small water bodies, mainly polygonal ponds, and small thermokarst and oxbow lakes. On the third terrace, polygonal ponds are also common, and large thermokarst lakes are usually situated in partially drained deep basins (locally called 'alasses').

### 2.2. Types of Waterbodies

Totally, 31 reservoirs typical of polygonal tundra were studied. On Kurungnakh Island, samples were collected in three thermokarst lakes on the first river terrace and ten

thermokarst lakes on the third terrace, as well as in three polygonal ponds on the first terrace and ten ponds on the third terrace. On Argaa-Bilir-Aryata Island, four polygonal ponds and one oxbow on the first river terrace were studied. Both in 2017 and 2020, surveys were conducted in two thermokarst lakes (Figure 1c).

*Thermokarst lakes.* Lakes on the first geomorphological terrace had flat boggy shores and a surface area of $4.8 \pm 3.7$ ha. Almost all the lakes on the third terrace were located in the alasses (Figure 2), and only two had flat shores without kettle depressions. One of the lakes sometimes partly dried. The lakes of the third terrace were mostly larger than those of the first terrace. Six had an area of $5.0 \pm 4.3$ ha, and two had an area of 48 ha. On the first river terrace, lakes had a maximum depth of approximately 2 m, and on the third terrace, lake depth was usually 3 m or more.

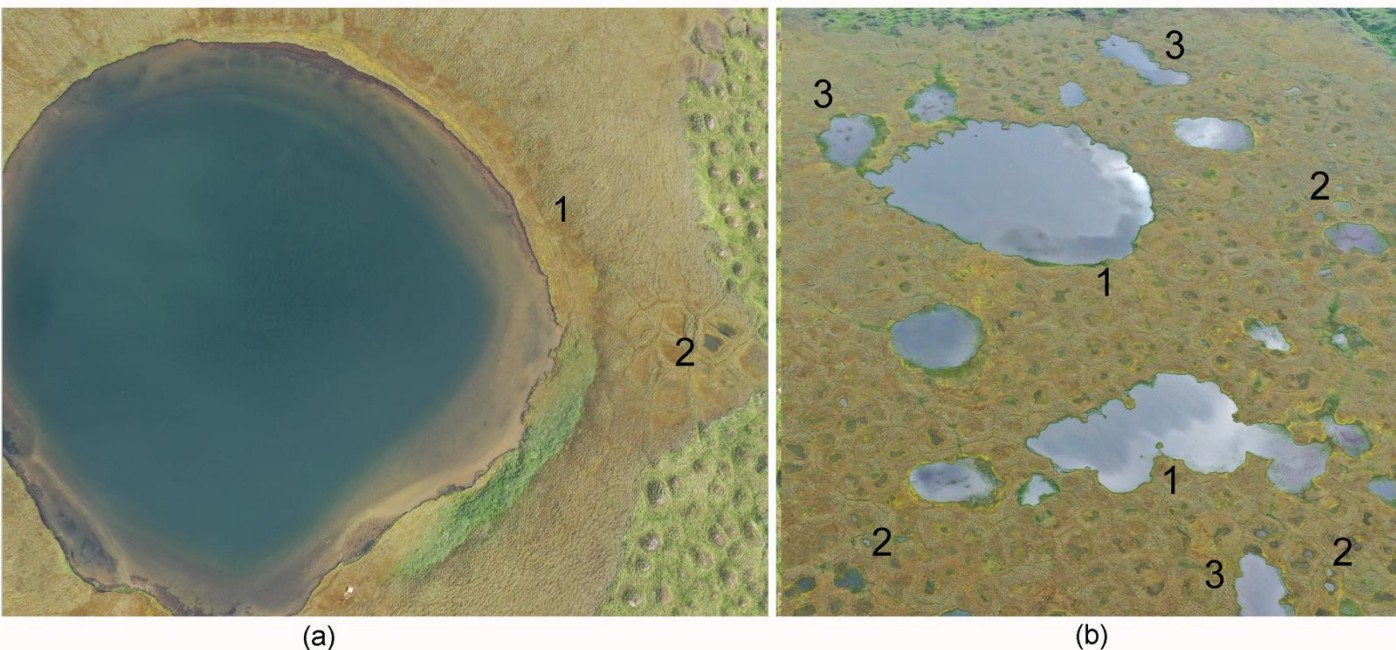

**Figure 2.** Main types of water bodies in Lena River Delta. Thermocarst lakes (1) located in allasse (**a**) and on flat tundra (**b**). Polygonal ponds: single (2) and complex (3).

Such differences in the morphology of thermokarst lakes on different river terraces are typical for the Lena Delta [22]. The bottoms of the lakes of both terraces were composed of fine sand with an admixture of silt and detritus. The permafrost near the shore was at a depth of 50–90 cm from the surface of the bottom sediments. The total mineralization of water in the lakes was $36.6 \pm 15.7$ ppm. Macrophyte thickets were confined to the lake margins and usually consisted of *Arctophila* Andersson (1852) and *Carex* L., 1753. During the research, numerous juveniles of *Pungitius pungitius* Linnaeus, 1758, and *Coregonus peled* (Gmelin, 1788) were noted in all thermokarst lakes.

*Polygonal ponds.* We divided the studied polygonal ponds into two groups: nine water bodies were single ponds, and eight ones were complex polygonal ponds (Figure 2a,b). Single polygonal ponds occupied depressions in the centre of one polygon that formed as a result of cryogenic processes in the active layer above the permafrost [26]. Complex polygonal ponds were 4–15 single ponds combined into one water surface. The average single polygonal pond area was $79.4 \pm 21.4$ m$^2$, and the average complex polygonal pond area was $691 \pm 331$ m$^2$. The depths of the studied ponds of both types did not exceed 1.5 m, and the bottom sediments consisted of clay silt and detritus. Permafrost near the shore of polygonal ponds laid at a depth of 25–55 cm from the surface of the bottom soil. The total mineralization of water in reservoirs was $34.1 \pm 18.3$ ppm. Single polygonal ponds were often either partially or fully covered with macrophytes, and complex ponds always had a free central part of the surface. Among macrophytes, *Arctophyla*, *Carex*, *Eriophorum*

L., 1753 and *Hippuris vulgaris* L., 1753 were dominant. Fish were not found in all types of polygonal ponds. The exception was only one flowing polygonal pond, in which three large individuals of *C. peled* were noted.

*Oxbows.* A single oxbow studied was located on Argaa-Bilir-Aryata Island and had an area of two ha. Its bottom was sandy, and the maximum depth was 1.5 m. Permafrost in the reservoir laid at a depth of 70 cm from the surface of the bottom soil. The total water mineralization (106 ppm) was higher than that in the thermokarst lakes and polygonal ponds. A similar specificity of the hydrochemical composition of the water in the oxbow was also noted for the island of Samoilovsky [27]. The strip of coastal macrophytes was formed by *Arctophyla*. In the shallow waters of the oxbow, numerous juveniles of *P. pungitius* were noted.

### 2.3. Sampling

At each site, quantitative samples of zooplankton were collected by hauling a plankton net (diameter 0.1 m, 50 μm mesh) horizontally through the water column parallel to the bottom. The volume of the filtered water was calculated based on the length of the net path through the water, measured at each site. Three replicates were taken at each station and combined into one mixed sample afterwards. The volume of each mixed sample was 48–50 L. The meiobenthos was sampled using a plastic tube that was inserted into the uppermost 3–4 cm of the sediment layer. From each site, three substrate portions were taken randomly, all representing different meiobenthic habitat substrates if possible, and then pooled. Each mixed sample covered an area of 9.4 $cm^2$. The samples were preserved with 96% ethanol and filtered (50-μm mesh) before identification. All the samplings were performed from the shore.

At each station, environmental variables such as water temperature, pH, and total mineralization (ppm) were measured with a Yieryi portable multifunctional electronic water quality tester (five in one). The depth of the permafrost layer in the coastal zone of the water bodies was measured with a specialized rod.

Preliminary species identification and counts were carried out in Bogorov counting chambers. The total numbers of Cladocera, Copepoda, Anomopoda, Ostracoda and Amphipoda were recorded. Copepodid stages of Cyclopoida and Calanoida were counted separately but only to the genus level without species identification. An Olympus CX-41 high-power microscope (Olympus Medical Systems Corporation, Tokio, Japan) was used for accurate crustacean identification following both standard taxonomic treatises and recent taxonomic revisions: 4, [28–33] for Copepoda; [34–40] for Cladocera; and [28] for Anomopoda; [7] for Amphipoda.

### 2.4. Statistical Analysis

To evaluate the effects of environmental factors on the crustacean community, we used distance-based linear modelling (DistLM) and PERMANOVA tests in PRIMER [41]. The first test was used to estimate the influence of environmental factors on species richness and general abundance in the observed water bodies, and the second test was applied to the species structure analysis. The environmental data involved 12 variables: TYPE—type of the water body; ISLAND—location of the sampling site at one of the investigated islands; YEAR—year of research; TERRACE—location of the sampling site on one of the investigated terraces; TEMP—temperature of water, °C; PPM—total mineralization; PH—pH; MACR—dominant macrophyte species in the water body; SUBSTR—type of bottom sediments; AREA—total area of the water body, $m^2$, log-transformed; DEPTH—average depth of the water body, m; and PERM—depth of permafrost, m. Data on the macrophyte composition were coded as presence/absence of dominant species and applied as group variable.

First, marginal tests were performed to determine the effect of each variable on the variation in species assemblage structure. Then, the best-fitting model was selected using the Akaike information criterion, AICc. This criterion was used to select significant factors in the model, taking into account sample size by increasing the relative penalty for model

complexity with small data sets. Sequential tests are provided for each variable that is added to the model.

The significance of the differences in the species composition and general abundance of the community fauna between years of research was separately analysed with the Mann–Whitney nonparametric test for medians. This test compares two sets of data, relying on the trend in the central values, and applies the non-Gaussian normal distribution of variables.

We also applied a constrained ordination technique, the canonical correspondence analysis (CCA), to determine the impact of the environmental variables on the invertebrate community and show the variations in the assemblages of organisms in accordance with the observed environmental factors in PAST [42]. For the DistLM analysis, data on the macrophyte composition was coded on the basis of the dominant species and analyzed as a group variable, area factor was log-transformed, while others were numerical.

## 3. Results

### 3.1. Species Richness and Abundance

Fifty-nine crustacean species and taxa were identified: 39 copepod species (belonging to 20 genera), 18 branchiopod species (13 genera), one Amphipoda species and no identified Ostracoda (Table 1). Five of these crustaceans had not previously been recorded from the Lena River Delta and neighbouring territories of NW Siberia, although they are quite widespread through the northern Palaearctic: two cladocerans (*Alona quadrangularis* (Müller O.F., 1776) and *Paralona pigra* Sars G.O., 1862) and three harpacticoids (*Attheyella* cf. *trispinosa* (Brady, 1880), *Bryocamptus arcticus* (Lilljeborg, 1902) and the typically brackish water *Nannopus procerus* Fiers and Kotwicki, 2013).

**Table 1.** Species list and presence of crustaceans from plankton and meiobenthos in water bodies of Kurungnakh and Argaa-Bilir-Aryata Islands (southern part of the Lena River Delta) in July–August 2017 and 2020 (*–species noted for the first time).

| Taxa | Number of Waterbodies | Lakes | Complex Polygon Ponds | Single Polygon Ponds | Oxbow |
|---|---|---|---|---|---|
| Class Malacostraca | | | | | |
| Order Amphipoda | | | | | |
| Family Crangonyctidae Bousfield, 1973 | | | | | |
| *Eosynurella jakutana* (Martynov, 1931) | 10 | + | + | + | - |
| Class Branchiopoda Order Anostraca | | | | | |
| Family Chirocephalidae Daday, 1910 | | | | | |
| *Polyartemia forcipata* (Fischer, 1851) | 3 | + | - | - | - |
| Family Branchinectidae Daday, 1910 | | | | | |
| *Branchinecta paludosa* (O.F. Müller, 1788) | 16 | + | + | + | - |
| Superorder Cladocera | | | | | |
| Order Ctenopoda | | | | | |
| Family Holopedidae G.O. Sars, 1865 | | | | | |
| *Holopedium gibberum* Zaddach, 1855 | 1 | + | - | - | - |
| Family Sididae Baird, 1850 | | | | | |
| *Sida crystallina* (O.F. Müller, 1776) | 3 | + | - | - | - |

**Table 1.** *Cont.*

| Taxa | Number of Waterbodies | Lakes | Complex Polygon Ponds | Single Polygon Ponds | Oxbow |
|---|---|---|---|---|---|
| | | Order Anomopoda | | | |
| | | Family Bosminidae Baird, 1845 | | | |
| *Bosmina longirostris* (O.F. Müller, 1785) | 8 | + | + | - | + |
| *Bo.* cf. *longispina* (O.F. Müller, 1785) | 4 | + | + | + | - |
| | | Family Chydoridae Dybowski and Grochowski, 1894 | | | |
| *Acroperus harpae* (Baird, 1834) | 7 | + | + | + | + |
| *Alona guttata* G.O. Sars, 1862 | 10 | + | + | + | - |
| * *Al. quadrangularis* (O.F. Müller, 1776) | 6 | + | + | + | + |
| *Alonopsis elongatus* G.O. Sars, 1862 | 20 | + | + | + | - |
| *Biapertura affinis* (Leydig, 1860) | 10 | + | + | + | - |
| *Chydorus* cf. *sphaericus* (O.F. Müller, 1785) | 27 | + | + | + | - |
| **Paralona pigra* G.O. Sars, 1862 | 2 | + | - | - | - |
| *Pleuroxus* cf. *trigonellus* (O.F. Müller, 1776) | 5 | + | - | + | - |
| | | Family Daphniidae Straus, 1820 | | | |
| *Daphnia cucullata* G.O. Sars, 1862 | 3 | + | + | - | - |
| *D.* cf. *longispina* (O.F. Müller, 1776) | 15 | + | + | + | + |
| *D.* cf. *pulex* Leydig, 1860 | 6 | + | + | + | - |
| | | Family Eurycercidae Kurz, 1875 | | | |
| *Eurycercus lamellatus* (O.F. Müller, 1776) | 5 | + | + | + | + |
| | | Class Hexanauplia | | | |
| | | Subclass Copepoda | | | |
| | | Order Calanoida | | | |
| | | Family Temoridae Giesbrecht, 1893 | | | |
| *Eurytemora gracilicauda* Akatova, 1949 | 1 | - | - | - | + |
| *Eur. gracilis* (G.O. Sars, 1863) | 6 | + | + | - | - |
| *Eur.* cf. *raboti* Richard, 1897 | 1 | - | + | - | - |
| *Heterocope borealis* (Fischer, 1851) | 24 | + | + | + | + |
| | | Family Diaptomidae Baird, 1850 | | | |
| *Eudiaptomus graciloides* (Lilljeborg, 1888) | 9 | + | + | + | - |
| *Leptodiaptomus angustilobius* (G.O. Sars, 1898) | 17 | + | + | + | - |
| *Mixodiaptomus theeli* (Lilljeborg in Guerne et Richard, 1889) | 16 | + | + | + | + |
| | | Order Cyclopoida | | | |
| | | Family Cyclopidae Rafinesque, 1815 | | | |
| *Acanthocyclops venustus* (Norman and Scott, 1906) | 25 | + | + | + | + |
| *Ac. vernalis vernalis* (Fischer, 1853) | 13 | + | + | + | - |
| *Cyclops scutifer scutifer* G.O.Sars, 1863 | 13 | + | + | + | - |
| *C. kolensis* Lilljeborg, 1901 | 19 | + | + | + | + |
| *C.* cf. *strenuus* Fischer, 1851 | 16 | + | + | + | + |
| *Diacyclops bicuspidatus* (Claus, 1857) | 6 | + | + | + | - |
| *Di. crassicaudis* (G.O. Sars, 1863) | 8 | + | - | + | + |
| *Di. languidoides* (Lilljeborg, 1901) | 13 | + | + | + | + |
| *Di. nanus* (G.O. Sars, 1863) | 2 | + | + | - | - |
| *Eucyclops* gr. *serrulatus* (Fischer, 1851) | 8 | + | + | + | - |

**Table 1.** *Cont.*

| Taxa | Number of Waterbodies | Lakes | Complex Polygon Ponds | Single Polygon Ponds | Oxbow |
|---|---|---|---|---|---|
| Family Cyclopidae Rafinesque, 1815 | | | | | |
| *Megacyclops gigas gigas* (Claus, 1857) | 10 | + | + | + | - |
| *Me. viridis viridis* (Jurine, 1820) | 13 | + | + | + | - |
| *Mesocyclops leuckarti* (Claus, 1857) | 2 | + | - | - | - |
| *Paracyclops fimbriatus* (Fischer, 1853) | 7 | + | + | + | + |
| Order Harpacticoida | | | | | |
| Family Canthocamptidae Brady, 1880 | | | | | |
| *Attheyella dentata* (Poggenpool, 1874) | 3 | + | + | + | - |
| *At. nordenskioldii* (Lilljeborg, 1902) | 1 | - | - | + | - |
| * *At.* cf. *trispinosa* (Brady, 1880) | 1 | - | - | + | - |
| * *Bryocamptus arcticus* (Lilljeborg, 1902) | 1 | + | - | - | - |
| *Br. vejdovskyi* (Mrazek, 1893) | 3 | + | - | + | - |
| *Br.* sp. 1 | 2 | + | - | - | - |
| *Br.* sp. 2 | 10 | + | + | + | + |
| *Canthocamptus glacialis* (Lilljeborg, 1902) | 17 | + | + | + | - |
| *Epactophanes richardi* Mrazek, 1893 | 2 | + | - | - | - |
| *Maraenobiotus brucei* (Ricard, 1898) | 6 | + | + | + | - |
| *Moraria duthiei* (Scott, 1896) | 18 | + | + | + | - |
| *Mo. insularis* Fefilova, 2008 | 3 | + | - | - | - |
| *Mo. mrazeki* Scott, 1903 | 20 | + | + | + | - |
| *M.* sp. | 13 | + | + | - | - |
| *Pesceus reductus* (M.S. Wilson, 1956) | 4 | + | + | - | - |
| *Pe. schmeili* (Mrazek, 1893) | 4 | + | - | - | - |
| *Pe.* cf. *reggiae* (M.S. Wilson, 1958) | 7 | + | + | - | - |
| Family Nannopodidae Brady, 1880 | | | | | |
| * *Nannopus procerus* Fiers and Kotwicki, 2013 | 1 | - | + | - | - |
| Class Ostracoda | | | | | |
| *Ostracoda* spp. | 7 | + | + | + | + |
| Total species richness: | | 54 | 43 | 39 | 16 |

Crustacean diversity in the studied water bodies was high: 16.6 species on average (ranging from 7 to 25). At the same time, the average number of crustaceans in one water body in the plankton was 13.03 ± 4.04 species, and that in the meiobenthos was 6.4 ± 3.4. The most common species in the studied water bodies were the Cladocera *Chydorus* cf. *sphaericus* (O.F. Müller, 1785) and the Copepoda *Heterocope borealis* (Fischer, 1851) and *Acanthocyclops venustus* (Norman and Scott, 1906). They each occurred in more than 23 localities (77–87%). The species *Branchinecta paludosa* (O.F. Müller, 1788); *Alonopsis elongatus* G.O. Sars, 1862; *Leptodiaptomus angustilobius* (G.O. Sars, 1898); *Mixodiaptomus theeli* (Lilljeborg in Guerne et Richard, 1889); *Cyclops kolensis* Lilljeborg, 1901; *C.* cf. *strenuus* Fischer, 1851; *Canthocamptus glacialis* (Lilljeborg, 1902); *Moraria duthiei* (Scott, 1896); and *M. mrazeki* Scott, 1903, were also quite frequent in the samples and occurred in 52–68% of the investigated water bodies. Approximately one-third of the species (22) were rare and occurred only in 1–5 water bodies. The highest number of species was observed in large thermokarst lakes (54 species), slightly fewer occurred in complex polygonal ponds (43 species), and only 39 occurred in single polygonal ponds. There was only one oxbow studied. Its species richness was rather low (16 species); however, the species *Eurytemora gracilicauda* Akatova, 1949, was observed only there.

It is notable, that the species richness decreased (by 15–30%) from large lakes to small polygonal ponds, and this trend was observed among almost all taxonomic groups of crustaceans: Anostraca, Ctenopoda, Anomopoda, Calanoida, Cyclopoida and Harpacticoida.

Ctenopoda were completely absent in small reservoirs. Among the anostracans, *Polyartemia forcipata* (Fischer, 1851) was found only in large lakes, and *Branchinecta paludosa* (O.F. Müller, 1788) inhabited various hydrological types of reservoirs. Amphipoda (*Eosynurella jakutana* (Martynov, 1931)) and Ostracoda were found in both lakes and small polygonal ponds.

The most abundant planktonic species were *Bosmina* cf. *longispina* (O.F. Müller, 1785), *L. angustilobius* and *M. theeli* (average 3.4–4.9 ind/L to 154 ind/L), and the most abundant meiobenthos were *Ca. glacialis*, *Mo. duthiei* and *Mo. mrazeki* (average 10.8–16.5 ind/10 cm$^2$ to 202 ind./10 cm$^2$). The total abundance of both planktonic and benthic crustaceans varied significantly even among reservoirs of the same hydrological type. However, there was a tendency of the abundance to increase along the gradient from large thermokarst lakes to single polygonal ponds. Thus, the number of crustaceans in the plankton and meiobenthos of thermokarst lakes was 42.3 ± 30.8 ind/L and 55.4 ± 61.8 ind/10 cm$^2$, respectively. In small polygonal ponds, these crustacean abundance values were much higher: 50.0 ± 53.7 ind/L in the plankton and 108.4 ± 159.4 ind/10 cm$^2$ in the meiobenthos.

*3.2. Patterns in Species Richness of Different Crustacean Groups*

The DistLM analysis showed that general species richness and the number of species from different groups of crustaceans chiefly depended on the year of research. Generally, the number of species was higher in 2017 than in 2020. At the same time, this factor depended on different forces on the two main groups, Cladocera and Copepoda. The statistical significance (p) of the differences between years, shown by the Mann–Whitney nonparametric test, was 0.0005 for the whole species list, 0.0036 for cladocerans, 0.02 for harpacticoid copepods and 0.0345 for non-harpacticoid copepods (cyclopoids and calanoids). This factor determined 38% of the differences in total species richness (27% of Cladocera, 16% of non-harpacticoid Copepoda and only 6% of Harpacticoida) (Table 2).

**Table 2.** The results of sequential test of DistLM (AIC criterion, step-wise selection). Significant factors are in bold ($p < 0.02$).

| Group | AIC | P | Prop. | Cumul. |
|---|---|---|---|---|
| | | Total number of species | | |
| +YEAR | 91.218 | 0.001 | 0.37953 | **0.37953** |
| +PH | 86.279 | 0.012 | 0.11767 | 0.4972 |
| | | Cladocera number of species | | |
| +YEAR | 26.473 | 0.004 | 0.27455 | **0.27455** |
| +TEMP | 23.819 | 0.044 | 0.09541 | 0.36996 |
| +PH | 20.342 | 0.023 | 0.09637 | 0.46633 |
| | | Non-harpacticoid Copepoda number of species | | |
| +YEAR | 59.766 | 0.018 | 0.15939 | **0.15939** |
| +PH | 54.685 | 0.009 | 0.16235 | **0.32173** |
| +DEPTH | 53.668 | 0.103 | 0.05925 | 0.38099 |
| +AREA | 51.12 | 0.057 | 0.0797 | 0.46069 |
| +PPM | 44.792 | 0.01 | 0.12029 | **0.58097** |
| +ISLAND | 43.354 | 0.109 | 0.041455 | 0.62243 |
| | | Harpacticoida number of species | | |
| +MACR | 47.234 | 0.01 | 0.36699 | **0.36699** |
| +YEAR | 45.828 | 0.118 | 0.062083 | 0.42908 |
| | | Cladocera/Copepoda number of species | | |
| +TEMP | −101.52 | 0.016 | 0.1722 | **0.1722** |

To a lesser degree, the number of Cladocera and non-harpacticoid Copepoda significantly depended on hydrochemical factors, pH and mineralization, and Cladocera was also influenced by temperature; however, macrophyte composition explained the highest amount (37%) of the variation in Harpacticoida species richness (Table 2).

The Cladocera/Copepoda species ratio slightly depended on temperature (Table 2). However, this relation was significant (r = 0.42, *p* = 0.016, Figure 3). The warmer the plot was, the higher the percentage of Cladocera species. The effect of low temperatures was much stronger on cladocerans than on copepods. In the current research, the only case where Cladocera noticeably dominated Copepoda in number was at the station with the highest temperature (18.7 °C).

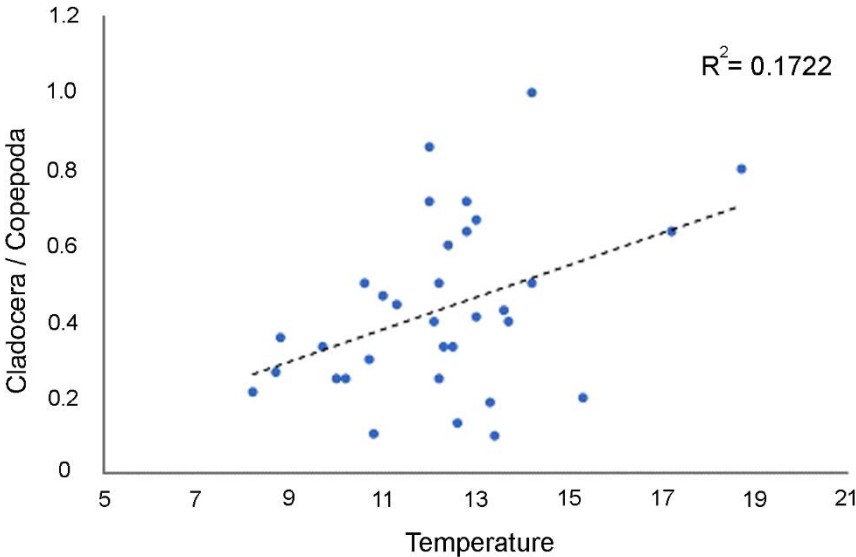

**Figure 3.** Relationship between temperature of water, measured at sampling sites, and the Cladocera/Copepoda ratio.

### 3.3. Variations in Species Composition and Assemblage Structure

According to PERMANOVA, various types of water bodies (large lakes or small ponds) differed from each other significantly only in terms of meiobenthic organisms, while others did not show any significant differences. The type of water body in this case reflects a bundle of related environmental factors. However, the method of constrained ordination (CCA) showed the specific variation in the species assemblages of hydrobionts in accordance with the different environmental factors. Both meiobenthic and planktonic organisms demonstrate matching patterns on the CCA plots (Figure 4). All samples are clearly arranged along the main axis of ordination, which includes correlated morphometric parameters of reservoirs (area, depth, permafrost depth, type of bottom sediments and composition of macrophytes). The first canonical axis is enough to represent the positional relationship of the samples and species. This axis describes almost 100% of the variation in the type of structure (significance of non-random axis allocation *p* = 0.001).

In the case of benthic organisms, at one end of the axis, large lakes are grouped (Figure 4). They are characterized by areas greater than 1000 m$^2$, depths of 2–3 m, permafrost depths greater than 0.5 m, sand-silt or sand bottom sediments and *Arctophyla* sp. dominating among macrophytes (sometimes together with *Carex* sp.). The species *Epactophanes richardi* Mrazek, 1893, *Pesceus schmeili* (Mrazek, 1893), *Pe.* cf. *reggiae* (Wilson, 1958), *Pe. reductus* (Wilson, 1956), *Br. arcticus*, *Br.* sp. 1, *Moraria insularis* Fefilova, 2008, *Mo. duthiei*, *Diacyclops crassicaudis* (G.O. Sars, 1863), *Biapertura affinis* (Leydig, 1860), *Bosmina longirostris* (O.F. Müller, 1785) and *Alo. elongatus* are typical here, comprising up to 100% (always more than half) of general abundance. At the other end of the axis, small ponds occur with areas less than 500 m$^2$, depths less than 1.5 m, permafrost located shallower than 0.5 m, silt or detritus at the bottom and variability in macrophytes often with the dominance of *Eriophorus* and *Hippuris vulgaris* Linnaeus, 1753. The species *Ca. glacialis*, *Attheyella nordenskioldii* (Lilljeborg, 1902), *At.* cf. *trispinosa*, *Bryocamptus vejdovskyi* (Mrazek, 1893), *Megacyclops viridis* (Jurine, 1820), *C. strenuus*, *Acroperus harpae* (Baird, 1834), *Ch. sphaericus*, and *D.* cf. *pulex* Leydig, 1860 and Ostracoda are common here.

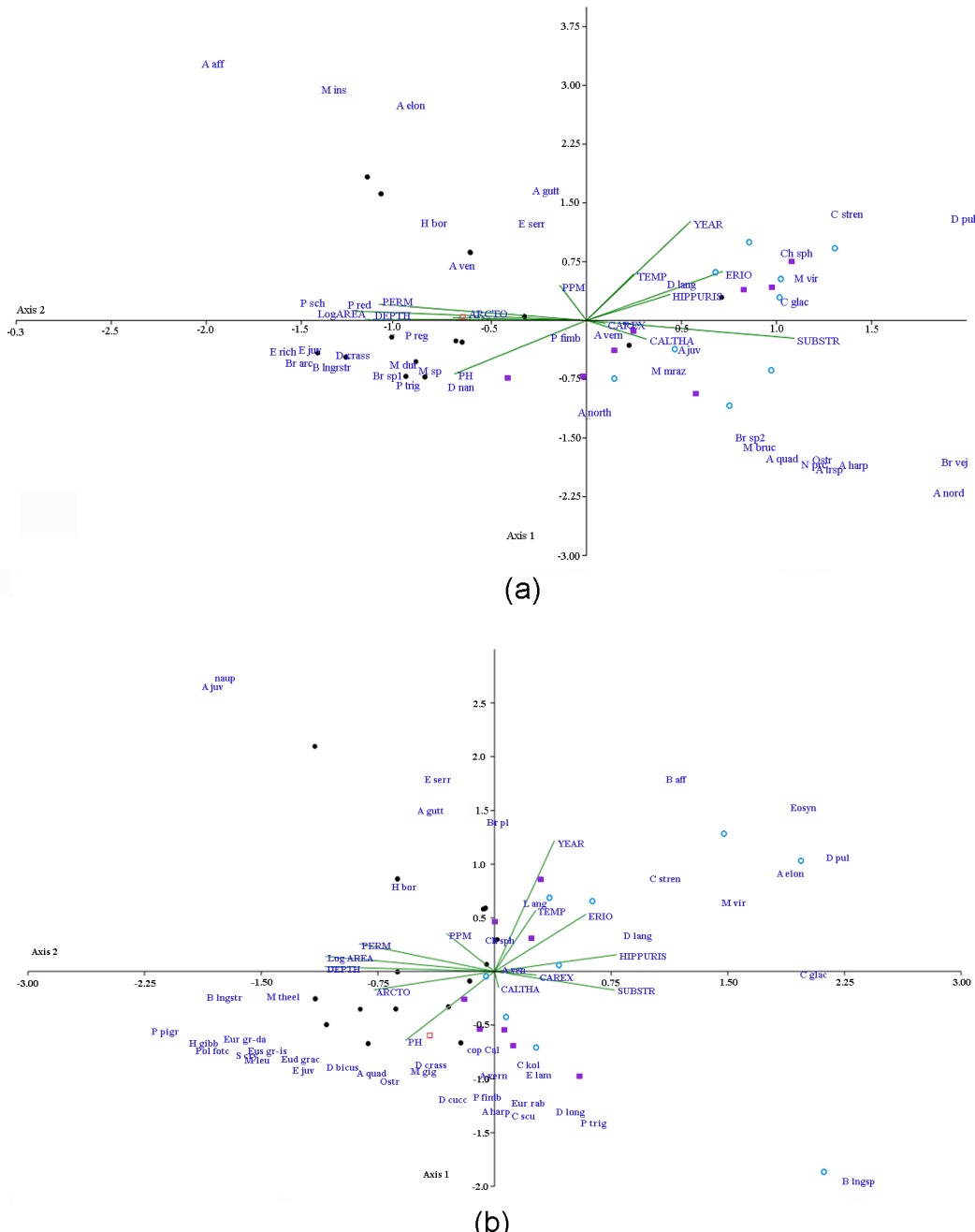

**Figure 4.** CCA ordination of benthic (**a**) and planktonic (**b**) crustaceans in the surveyed sites of the Lena Delta. Environmental variables included in both plots. Species are abbreviated as: A elon-*Alo. elongatus*; A gutt-*Al. guttata*; A harp-*A. harpae*; A nord-*At. nordenskioldii*; A den-*At. dentata*; A juv-Aloninae juv; A quad-*Al. quadrangularis*; A trsp-*At.* cf. *trispinosa*; A ven-*Ac. venustus*; A vern-*Ac. vernalis*; B aff-*Bi. affinis*; B lngsp-*Bo.* cf. *longispina*; B lngrstr-*Bo. longirostris*; Br arc-*Br. arcticus*; Br pal-*B. paludosa*; Br sp1-*Br.* sp. 1; Br sp2-*Br.* sp. 2; Br vej-*Br. vejdovskyi*; C glac-*Ca. glacialis*; Ch sph-*Ch.* cf. *sphaericus*; C scu-*C. scutifer*; C stren-*C.* cf. *strenuus*; C kol-*C. kolensis*; Cop Cal-copepodid Calanoida; D bicus-*Di. bicuspidatus*; D crass-*Di. crassicaudis*; D cuc-*D. cucullata*; D lang-*Di. languidoides*; D long-*D.* cf. *longispina*; D nan-*Di. nanus*; D pul-*D.* cf. *pulex*; E lam-*Eu. lamellatus*; E rich-*Ep. richardi*; E ser-*Euc.* gr. *serrulatus*; Eosyn-*E. jakutana*; E juv-*Eurycercus* juv; Eud grac-*Eud. graciloides*; Eur gr-da-*Eur. gracilicauda*; Eur gr-is-*Eur. gracilis*; Eur rab-*Eur.* cf. *raboti*; H bor-*He. borealis*; H gibb-*H. gibberum*; L ang-*L. angustilobius*; M bryc-*Ma. brucei*; M dut-*Mo. duthiei*; M gig-*Me. gigas*; M ins-*Mo. insularis*; M leu-*Mes. leuckarti*; M mraz-*Mo. mrazeki*; M sp-Mo. sp.; M theel-*M. theeli*; M vir-*Me. viridis*; N pro-*N. procerus*; naup-nauplii Copepoda; Ostr-Ostracoda spp.; P fimb-*Par. fimbriatus*; P pigr-*Pa. pigra*; P sch-*Pe. schmeili*; P red-*Pe. reductus*; P reg-*Pe.* cf. *reggiae*; P trig-*Pl.* cf. *trigonellus*; Pol forc-*P. forcipata*; S cry-*S. crystallina*. Colored symbols are water bodies of different types: black dots-big lakes, violet squares-complex ponds, blue circles-small ponds, red square-oxbow.

Analysis of planktonic organisms revealed very similar patterns of distribution in the CCA plot (Figure 4). All samples are also clearly lined along the main axis of ordination, including correlated morphometric parameters of water bodies. In addition, all the sites can be divided into three groups:

*Group A* (left part of the first axis of ordination)—this group has the same characteristics as those of the benthic group: large lakes with areas greater than 10,000 m$^2$ and more than 2 m in depth, permafrost depths more than 0.5 m, sand-silt or sand bottom sediments and *Arctophyla* sp. dominating among macrophytes (sometimes together with *Carex* sp.). The typical species are *Polyartemia forcipata* (Fischer, 1851), *Sida crystallina* (O.F. Müller, 1776), *Holopedium gibberum* Zaddach, 1855, *Bo. longirostris*, *Alona guttata* G.O. Sars, 1862, *Al. quadrangularis*, *Eurytemora gracilicauda* Akatova, 1949, *Eur. gracilis* (G.O. Sars, 1863), *He. borealis*, *Eudiaptomus graciloides* (Lilljeborg, 1888), *M. theeli*, *Mesocyclops leuckarti* (Claus, 1857), *Diacyclops bicuspidatus* (Claus, 1857), *Eucyclops* gr. *serrulatus* (Fischer, 1851), and Ostracoda.

*Group B* (medium part of the first axis of ordination): this group contains water bodies of various types with areas of 600–48,000 m$^2$, depths of 0.5–1.5 m, permafrost depths of 0.3–0.9 m, different sediments from sand to silt with detritus and *Carex* sp. dominating among macrophytes (sometimes together with *Arctophyla* sp.). The common species are *B. paludosa*, *A. harpae*, *Ch. sphaericus*, *Daphnia cucullata* G.O. Sars, 1862, *Di. crassicaudis*, *C. kolensis*, *Paracyclops fimbriatus* (Fischer, 1853), *Megacyclops gigas* (Claus, 1857), *Acanthocyclops vernalis* (Fischer, 1853), *Ac. venustus*, and *L. angustilobius*.

*Group C* (right part of the first axis of ordination): this group includes small shallow (depth < 0.5 m) ponds with areas less than 600 m$^2$ and permafrost depths less than 0.5 m. The sediment is silt and detritus, and there are various macrophytes: *Carex* sp., *Hippuris vulgaris*, *Eriophorum* sp. Typical species: *Bi. affinis*, *Alo. elongatus*, *Pleuroxus* cf. *trigonellus* (O.F. Müller, 1776), *D.* cf. *longispina* (O.F. Müller, 1776), *D.* cf. *pulex*, *Bo. longispina*, *Eurycercus lamellatus* (O.F. Müller, 1776), *Eurytemora* cf. *raboti* Richard, 1897, *Cyclops scutifer* G.O.Sars, 1863, *Diacyclops languidoides* (Lilljeborg, 1901), *C. strenuus*, *Me. viridis*, *Ca. glacialis*, and Crangonyctidae.

Species of the dominant group account an average of 58–76% of the total abundance (Table 3). However, there was no clear gap between large lakes and small ponds, and the groups described above were not discrete. The sampling sites represent a smooth gradient along the ordination axis according to the morphometry of the water bodies from large lakes to small ponds.

**Table 3.** Summarized percentage of abundance for three groups of characteristic species in three groups of waterbodies: mean (min-max). Wb = water body.

|  | **Species Group A** | **Species Group B** | **Species Group C** |
| --- | --- | --- | --- |
| Wb Group A | 63 (46–81) | 31 (4–53) | 6 (0.3–15) |
| Wb Group B | 12 (0–31) | 76 (52–95) | 12 (2–48) |
| Wb Group C | 4 (0–17) | 38 (1–70) | 58 (27–99) |

Nonparametric distance-based multiple regression (DistLM) showed that the first CCA axis scores were used as independent variables but were highly significant ($p$ = 0.001) and explained only 21% of the total differences in benthic assemblages and 8% in planktonic assemblages. This value is low because most species had a low occurrence frequency. Seventeen out of 40 benthic species were found in one or two samples, and only two species occurred in more than half of the samples; out of 45 taxa, only 8 were found in more than half of the planktonic samples. This low occupancy resulted in high compositional variability and inflated the predictive power of the environmental variables in this type of analysis.

## 4. Discussion

### 4.1. Fauna and New Records for the Region

To date, the freshwater and brackish water Copepoda crustaceans of the Lena River Delta account for 76 species (18—Harpacticoida, 37—Cyclopoida, and 21—Calanoida) [21], and Cladocera accounts for 36 species (Anomopoda—30, Ctenopoda—5, and Onychopoda—1) [13–16,43]. Anostraca includes five species, Amphipoda includes two species, and Notostraca and Isopoda include one species each [8]. In the present study, two more species of Cladocera and three species of Copepoda were noted for the first time in the area. Their distribution and ecology are discussed in detail below.

Cladocera. The species *Alona quadrangularis* (Müller O.F., 1776) and *Paralona pigra* Sars G.O., 1862, typical of boggy water bodies (pH < 6) with silty bottom sediment, were found [28]. Of them, *Al. quadrangularis* is widespread in the Holarctic and known from many subarctic and Arctic regions, both western Siberia and the Far East (Chukchi Peninsula). *Pa. pigra* is cosmopolitan and found everywhere but Australia and Antarctica [40]. The species is rare at high latitudes and has been found on Bering Island [44] and in the Northwest Territories of Canada [45]. Copepoda. Three species of Harpacticoida are new for the Lena Delta. The only specimen of *Attheyella* cf. *trispinosa* (Brady, 1880) was found in a single oxbow on Argaa-Bilir-Aryata Island. The species is known from very distant locations—Western Europe and Northern Africa [29]. Possibly, despite the morphological likeness with the species *At. trispinosa*, this individual could belong to another undescribed species of the *Attheyella* genera. *Bryocamptus arcticus* (Lilljeborg, 1902) was observed in the thermokarst lake on Kurungnakh Island with silty and sandy bottom sediment. The distribution of the species is quite wide as it is found from the Bolshezemelskaya tundra to the Scandinavian Peninsula and in Greenland [33]. The southern boundary of the range of *Br. arcticus* apparently coincides with the border of sphagnum peatbogs [29]. Additionally, in the waters of the delta, the brackish water species *Nannopus procerus* Fiers and Kotwicki, 2013, characteristic of the littoral zone of the northern coast of Europe and the White Sea [46], as well as the coastal zone of the southern part of the Kara Sea [47], was found. In the Lena Delta, the only individual of this species was found in a complex pond with a total mineralization of only 17 ppm. Most likely, *N. procerus* was introduced to the reservoir from the plumage of ducks wandering from the outshore to the continental part of the delta. For all three Harpacticoida species, their location in the Lena River Delta is the easternmost point of their now known range.

The microcrustacean species that were discovered for the first time in the delta are not invaders from more southern regions and were not noted earlier due to lack of knowledge of the reservoirs. For example, *Pa. pigra* has small body shell sizes (<0.4 mm) and could be mistaken for juvenile individuals of other species of the Chydoridae family. Other newly found species are also relatively small (<0.8 mm) organisms, and their abundance and occurrence in reservoirs were low.

Several copepod species identified in the present study were previously listed as occurring in the Lena Delta in only a conference abstract [20], and their data are not in the publicly available literature. These species include *Diacyclops nanus* (Sars G.O., 1863), which has a wide Holarctic range and is often found at the bottom of delta reservoirs [33]. In addition to the typical species *Pe. schmeili* that occurs in many arctic regions, two additional species of this genus were noted in the lakes and complex polygonal ponds—*Pe. reductus* and *Pe.* cf. *reggiae*. The first species was previously known from the tundra reservoirs of Alaska and from Hokkaido Island [48]. The second species was before recorded only from Alaska. It is assumed that *Pe.* cf. *reggiae* is a new species [20].

In addition, it should be noted that Palearctic populations of *Bi. affinis* were recently split on the basis of morphological and genetic analysis, and now this species is a group of species [49]. It is possible that *Bi. affinis* found in the Lena Delta may actually refer to the closely related species *Biapertura sibirica* (Sinev, Karabanov, Kotov, 2020). However, this assumption has not been tested on our material.

Despite the existing opinion about the paucity of the microcrustacean arctic fauna [50], the Lena River Delta is distinguished by its high richness of the Cladocera and, especially, Copepoda compared to that in other Arctic regions at the same latitude. In recent studies, 11 new species of Copepoda (7 Harpacticoida and 4 Cyclopoida) have been discovered in the delta [20,21]. Most likely, the high diversity of crustaceans is high because during the ice age some refugia are located in this region. It is assumed that the crustacean fauna of the Lena River Delta is a fragment of the fauna of Beringia, the disappeared land, which included the territories of the Commander and Aleutian Islands and Alaska [51–53]. This fact indirectly confirms the presence of Copepoda species in the delta, with split ranges covering together eastern Siberia, Alaska and Japan. Phylogeographic studies confirm a special status of the Beringian zone and a specific history of the taxon dispersion there. As a result of the works on the cladoceran taxa distribution, the separation of two main biogeographic provinces (Western Holarctic Province and Beringian Province) of the Holarctic with a transitional zone in Eastern Siberia was shown [54–56]. Crustaceans species here can differ from that in other regions due to the differences in geological and climatic events during the last glacial maximum and previous Pleistocene glacial cycles.

### 4.2. Species Richness and Abundance

In the Lena River Delta, the species richness of crustaceans is higher in the thermokarst lakes than in the single polygonal ponds (Table 1). Although most lakes, unlike ponds, are inhabited by fish, this did not lead to a decrease in the number of species. The paucity of the fauna in the small reservoirs is obviously associated with their freezing to the bottom during the winter, which leads to the death of organisms with no persistent resting stages [19]. For example, representatives of Ctenopoda (Cladocera) that do not form ephippia with resting eggs [57] were completely absent in the single polygonal ponds of the delta. Species of this taxonomical group have only diapausing stages that could be more susceptible to frost. In contrast, the number of crustaceans was slightly higher in the small reservoirs than in the lakes. This fact is due to both zooplankton being eaten by fish in the large lakes and to the increased warming of small shallow ponds, which contributes to the active development of organisms [58].

### 4.3. Crustacean Assemblage Structure and Regulating Factors

The most significant influence on the structure of the crustacean assemblies was exerted by four groups of factors: the year of research, water temperature, hydrological and hydrochemical characteristics of the reservoir, and macrophyte composition (Table 2). Of these factors, the first two and second two were partially correlated with each other. For the first pair, this correlation occurred because when alternating warm and cold summer seasons, the heating temperature of the water masses varies. Therefore, the summer of 2017 was significantly colder than that of 2020, and the average water temperature in the lakes and polygonal ponds during these years differed by two degrees (11.5 and 13.5 °C, respectively). For the second pair, the correlation occurred because water bodies with hydrological differences have different hydrochemical characteristics, which, in turn, determine the formation of phytocenoses [58]. The variety in the water bodies of the Lena River Delta represents different stages of lake development. The characteristics of lakes, such as the depth of permafrost, general mineralization and water pH, gradually increase from single polygonal lakes to thermokarst lakes. Against the background of these changes, the phytocenosis of the water bodies changes. Thermokarst lakes of the delta were characterized by the dominance of species of *Arctophyla* and *Carex*. In the macrophyte community of the complex polygonal ponds, the proportion of *Eriophorum* increased, and *H. vulgaris* Linnaeus, 1753, and *Caltha* began to occur. Of the single polygonal pond species, *Eriophorum* and *H. vulgaris* were usually dominant.

Factors also differed in their strength of influence on the different taxa of Crustacea. Thus, the number of species and abundance of Cladocera were determined primarily by the year of research and the associated water temperature. Temperature had a smaller effect

on non-harpacticoid copepods than on Cladocera, while the importance of the substrate and hydrochemical characteristics of the reservoir increased. It had been shown previously that Cladocera is much more sensitive than Copepoda to low temperature [59–61]. When comparing microcrustacean faunas of different arctic regions, the nonlinear effect of temperature was shown. The regression models predicted that a one-degree drop in temperature, with all other things being equal, would lead to a 53% decrease in cladoceran richness on average, while for copepods, the corresponding decrease was only approximately 20% [60]. This difference in the diversity-temperature relationships resulted in compositional changes along the climatic gradient: copepods were dominant at temperatures below 13–15 °C, whereas cladocerans were dominant in warmer regions.

It is notable that Cladocera of the order Anomopoda form resting eggs, which are able to survive an unfavourable period for a long time [40]. Thus, in the bottom sediments of arctic reservoirs, there is constantly a hidden pool of species (including boreal), which can be introduced into the community when favourable climatic conditions arise [60]. The variability in the distribution of Harpacticoida in the reservoirs of the Lena Delta was mainly controlled by the composition of macrophytes in the reservoir, which distinguished them from other orders of Copepoda. The dependence of the harpacticoid distribution on the structure of aquatic phytocenosis has been repeatedly noted for marine ecosystems [62]. In freshwater, the role of macrophytes in assemblages of Harpacticoida remains poorly understood.

Assemblages of the Crustacea were not discrete in different hydrological types of lakes and ponds. Their structures change gradually, forming a continual gradient from single ponds to large thermokarst lakes (Figure 4). The variability in the assemblages was partially (more for Copepoda) associated with hydrology and hydrochemical characteristics that correlated with the age of the reservoir [58]. The general development cycle of tundra reservoirs forming on permafrost includes the following stages: a frosty mound; a polygon with an increasing degree of watering; merging neighbouring polygons; and a thermokarst lake [63,64]. Using the example of the water bodies in the Lena River Delta, one can describe the successional variability in crustacean assemblages corresponding to the main stages of this cycle. At the stage of single polygonal ponds, Cladocera are usually a dominant element in freshwater zooplankton communities (>60% of the total abundance): *D. longispina, D. pulex,* and *Bo. longispina*. In the complex polygonal pond stage, copepods prevail in plankton: *Ac. venustus, Ac. vernalis, C. kolensis* (Cyclopoida), *L. angustilobius* (Calanoida) and the small Cladocera *Ch. sphaericus*. The high abundance of large Calanoida is typical for the plankton in thermokarst lakes: *M. theeli, Eud. graciloides,* and *He. borealis*. In the meiobenthos of single polygonal lakes, the copepods *Ca. glacialis, Br. vejdovskyi* (Harpacticoida), *Me. viridis,* and *C. strenuus* (Cyclopoida) and the facultatively planktonic cladoceran *Ch. sphaericus* dominate, while in thermokarst lakes, only the Harpacticoida *Ep. richardi, Mo. insularis, Pe. reductus, Pe.* cf. *reggiae* and *Mo. duthiei* occur.

Apparently, the perennial variability in the meiobenthic assemblage structure is smoother than that in the planktonic assemblage structure. This scenario occurs because the planktonic fauna changes significantly under the influence of summer temperatures, varying over the years. The structure of the meiobenthic fauna reflects the stage of development of the reservoir and is associated primarily with the composition of phytocenosis.

The formation of crustacean assemblages in the Arctic reservoirs of the Lena River Delta is mainly determined by two types of ecological filters. For planktonic crustaceans, one or more faunal complex is formed on the basis of the hydrological characteristics and hydrochemistry of the water masses of a particular reservoir. The annual repeatability of this complex arises mainly due to the resting stages stored in the bottom sediments. The temperatures of the summer season determine which part of the hidden pool of the resting stages is uncovered each year. For meiobenthic crustaceans, local features of the reservoir, primarily the composition of phytocenosis, are also important. The influence of temperature on the meiobenthic assemblages within an individual summer season is not

significant, but the overall perennial temperature trend reflecting directed climate change is probably important.

Despite the high level of adaptation, freshwater ecosystems of Arctic are among the most sensitive to environmental change. Currently, polar regions, including the Lena Delta, face numerous stressors, such as the constant influx of pollutants, increased exposure to ultra-violet radiation, and, of course, climate change. In this regard, it is particularly important to pay attention to the study of their inland waters. Due to their peculiarities, freshwater zooplankton and meiobenthos communities of high latitudes are excellent model objects for environmental research in a changing climate.

## 5. Conclusions

1.  In the present study, 59 crustacean species and taxa were found in the water bodies of the southern Lena River Delta: 39 Copepoda, 16 Cladocera, 2 Anostraca, 1 Amphipoda and Ostracoda (not identified). Five of these crustaceans (*Al. quadrangularis, Pa. pigra, At.* cf. *trispinosa, Br. arcticus*, and *N. procerus*) are new for the region.
2.  The species richness of crustaceans was higher in the thermokarst lakes than in the single polygonal ponds due to the freezing of small reservoirs during the winter. In contrast, the abundance of crustaceans was lower in the lakes than in the shallow ponds, which warms more in summer and lacks fish.
3.  Variations in the Cladocera assemblage structure are due to annual differences in the water temperature (connected to the year of research) and, to a lesser degree, hydrochemical features of the water bodies. The structure of non-harpacticoid Copepoda was generally determined by hydrochemical factors and less affected by the year of research. The main factor that was sensitive to Harpacticoida was the composition of macrophytes.
4.  Three types of crustacean assemblages characteristic of different stages of the development of tundra water bodies were distinguished. At the single polygonal pond stage, large species of Cladocera were mostly dominant in the plankton; at the complex polygonal pond stage, the most abundant species were Copepoda in the Cyclopoida family; and in thermokarst lakes, large Calanoida copepods prevailed. The meiobenthic crustacean fauna of single polygonal lakes consisted of several Harpacticoida and Cyclopoida species together with the cladoceran *Ch. sphaericus*, while the complex of species in thermokarst lakes included only harpacticoids.

**Author Contributions:** Conceptualization E.S.C., A.A.N.; methodology E.S.C.; writing E.S.C., A.A.N., revision E.S.C., A.A.N. All authors have read and agreed to the published version of the manuscript.

**Funding:** The study of Cladocera was funded by the Russian Science Foundation, Grant Number 18-14-00325; study of Copepoda was funded by the Russian Foundation for Basis Research, Grant Number 20-04-00145.

**Institutional Review Board Statement:** Not applicable.

**Informed Consent Statement:** Not applicable.

**Data Availability Statement:** Not applicable.

**Acknowledgments:** The authors are grateful to the staff of the Lena Delta Nature Scientific Reserve and to the members of Russian–German Expeditions to Siberia in 2017 and 2020 for help in organising of the field works. Many thanks to Abramova, E.N., Praz, C., Sadchikov, I.P. and Vorobjeva, L.V. for their assistance during the sampling in the Lena River Delta. We are grateful to American Journal Experts (AJE) for English corrections by native speaking editors (certificate D7F5-85FE-BO7E-AA56-22C4).

**Conflicts of Interest:** The authors declare no conflict of interest. The funders had no role in the design of the study; in the collection, analyses, or interpretation of data; in the writing of the manuscript, or in the decision to publish the results.

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
