# Peer review of "Crustaceans in the Meiobenthos and Plankton of the Thermokarst Lakes and Polygonal Ponds in the Lena River Delta (Northern Yakutia, Russia): Species Composition and Factors Regulating Assemblage Structures"

_water, doi:10.3390/w13141936_

Round 1
Reviewer 1 Report
Review of the manuscript
Chertoprud E.S., Novichkova A.A. Crustaceans in the Meiobenthos and Plankton of the Thermokarst Lakes and Polygonal Ponds in the Lena River Delta (Northern Yakutia, Russia): Species Composition and Factors Regulating Assemblage Structures
This is an excellent and important ecological study on freshwater Crustaceans of the Arctic. The authors provide an exhaustive comparative analysis of the assemblages of several different crustacean major groups collected in thermokarst lakes, polygonal ponds, and one oxbow lake in the Lena delta. With regard to the content, I do not have any objection; the introduction provides comprehensive insights even to readers that are (like me) not familiar with freshwater studies and the study area. In the material and methods section, the authors explain quite understandable the applied methods and the used statistics. In the results section, these are presented detailed but very clear. The argumentation given in the discussion is comprehensive and well-founded. The conclusions are presented in a straight and summarizing form. The applied statistical methods are adequate, as I am concerned, and the used literature seems to be appropriate. The text is well-written, and I could follow the thoughts and argumentation of the authors without problems. Thus, I have no suggestions to make with respect to the content of the manuscript – congratulations to the authors for their excellent work!
However, I found some slips of the pen (marked in the uploaded revised draft) that should be corrected. Moreover, I would like to recommend modifying the abbreviation of genus names in the text. The authors deal with different crustacean taxa, and these include several different genera whose names, however, start with the same letter. Therefore, an abbreviation of the genus name by giving simply the first letter may confuse the reader, in particular if she/he is no expert in the respective major groups. For instance, Mixodiaptomus theeli (Calanoida), Megacyclops gigas gigas (Cyclopoida), and Moraria duthiei (Harpacticoida) should be abbreviated with Mi. theeli, Meg. gigas gigas, and Mo. duthiei, instead of M. theeli, M. gigas gigas, and M. duthiei. That procedure is becoming more and more asserted, and it is quite helpful for readers of papers dealing with large numbers of species and genera.
I suggest publication of the manuscript after minor revision.

Author Response
Answer on review 1
Thank you very much for your positive review of our article.
Answer on reviewer comments
>However, I found some slips of the pen (marked in the uploaded revised draft) that should be corrected.
Answer: All stylistically mistakes were corrected accordingly reviewer recommendations. All correction in the text of manuscript were accepted, except one (we write these below).
>To avoid confusion of the reader, I suggest replacing stop by colon (Thermokarst lakes:, Polygonal ponds:, Oxbows:). Lines: 112, 127, 145.
Answer: We didn’t use colons, but we did italicize these subheadings.
>Moreover, I would like to recommend modifying the abbreviation of genus names in the text. The authors deal with different crustacean taxa, and these include several different genera whose names, however, start with the same letter. Therefore, an abbreviation of the genus name by giving simply the first letter may confuse the reader, in particular if she/he is no expert in the respective major groups.
Answer: Abbreviation of Crustacean genus names were modified through the text, in Table 1 and in the caption to figure 4. Genera of species that stay first in Table 1 abbreviated to one letter, next genera (beginn from similar letter) - to two first letters, etc. For example, Holopedium gibberum - H. gibberum; Heterocope borealis - He. borealis. We don't change captions of points on Figure 4 because it will be more longer, and on figure place is limited.

Reviewer 2 Report
Review of a manuscript submitted to the Water, entitled “Crustaceans in the Meiobenthos and Plankton of the Thermokarst Lakes and Polygonal Ponds in the Lena River Delta (Northern Yakutia, Russia): Species Composition and Factors Regulating Assemblage Structures”, by Elena S. Chertoprud and Anna A. Novichkova.
Information about invertebrates in the low-flow water bodies of northeastern Siberia is far from complete. Traditionally, hydrobiological studies of the continental aquatic bodies are confined to Western Europe; North America; and, in Russia, to the northern half of its European part and south of the Far East. In particular, only fragmentary data on the composition of rheophilic fauna of the The Lena River Delta (and the overall tundra zone of Siberia) are currently available and there is almost no information about its ecology.
From the scientific point of view but also for the restoration practice the work might miss any novelty, however still I would strongly support a publication of this study. Such detailed and comprehensive information as presented here can be usually only obtained with a rather high effort (really nicely accomplished by the authors) and/or are sometimes even not accessible or citable. The objective of the study has been met, and research findings have been thoroughly discussed. The paper title match its contents, the key words and the abstract characterize the contents of the paper sufficiently, the objective of the paper formulated correctly, the material and research methods presented appropriately and clearly, the assumptions formulated in the objective been achieved, the data contained in tables and figures represent the appropriate, understandable documentation of the contents of the paper, the discussion of results correct and sufficient, the items of literature included in the paper sufficient and adequate to the subject of the paper, the conclusions formulated correctly and justified by the contents and results of the study.
Author Response
Answer on review 2
Thank you very much for your positive review of our article.
We have answer on yours comment: "From the scientific point of view but also for the restoration practice the work might miss any novelty ..."
Answer:
The novelty and the value of this study lie in the simultaneous analysis of many taxonomic and ecological (benthos and plankton) groups of crustaceans. The work includes a thorough and proper analysis of species taxonomy of all the taxa involved. At the same time, freshwater ecosystems of high latitudes are excellent model objects for research in many areas of science, as well as key sites for environmental research, monitoring and observation of global climate change. Such information were added in Introduction and Discussion.
Lines: 81-84; 535-541.
